# Overcoming Hypoxia-Induced Drug Resistance via Promotion of Drug Uptake and Reoxygenation by Acousto–Mechanical Oxygen Delivery

**DOI:** 10.3390/pharmaceutics14050902

**Published:** 2022-04-20

**Authors:** Yi-Ju Ho, Dinh Thi Thao, Chih-Kuang Yeh

**Affiliations:** 1Department of Biological Science and Technology, National Yang Ming Chiao Tung University, Hsinchu 30010, Taiwan; yijuho@nycu.edu.tw; 2Department of Biomedical Engineering and Environmental Sciences, National Tsing Hua University, No. 101, Section 2, Kuang-Fu Road, Hsinchu 30013, Taiwan; dinhthaonbhup@gmail.com

**Keywords:** hypoxia, drug resistance, reoxygenation, permeability, ultrasound

## Abstract

Hypoxia-induced drug resistance (HDR) is a critical issue in cancer therapy. The presence of hypoxic tumor cells impedes drug uptake and reduces the cytotoxicity of chemotherapeutic drugs, leading to HDR and increasing the probability of tumor recurrence and metastasis. Microbubbles, which are used as an ultrasound contrast agent and drug/gas carrier, can locally deliver drugs/gas and produce an acousto–mechanical effect to enhance cell permeability under ultrasound sonication. The present study applied oxygen-loaded microbubbles (OMBs) to evaluate the mechanisms of overcoming HDR via promotion of drug uptake and reoxygenation. A hypoxic mouse prostate tumor cell model was established by hypoxic incubation for 4 h. After OMB treatment, the permeability of HDR cells was enhanced by 23 ± 5% and doxorubicin uptake was increased by 11 ± 7%. The 61 ± 14% reoxygenation of HDR cells increased the cytotoxicity of doxorubicin from 18 ± 4% to 58 ± 6%. In combination treatment with OMB and doxorubicin, the relative contributions of uptake promotion and reoxygenation towards overcoming HDR were 11 ± 7% and 28 ± 10%, respectively. Our study demonstrated that reoxygenation of hypoxic conditions is a critical mechanism in the inhibition of HDR and enhancing the outcome of OMB treatment.

## 1. Introduction

Tumor hypoxia is a critical microenvironment that correlates with treatment resistance, recurrence, and metastasis. When the rate of tumor cell proliferation is faster than that of angiogenesis, the distance between vessels and cells is greater than the range of oxygen diffusion (>70 μm) [1]. Thus, tumor hypoxia occurs in a tumor microenvironment with insufficient oxygen supply. During hypoxia, tumor cells express hypoxia inducible factor-1 alpha (HIF-1α) to excessively activate vascular endothelial growth factor (VEGF) for promoting angiogenesis [2]. However, the imbalance between pro- and anti-angiogenic factors produces an immature and dysfunctional vasculature that further impedes blood perfusion within tumors [3]. Although numerous tumors reveal angiogenesis, the abnormal vasculature reduces oxygen and drug delivery to inhibit the efficacy of chemotherapy and radiotherapy. Moreover, the expression of HIF-1α also regulates the glycolytic metabolism, migration, and invasion in tumor cells to adapt to hypoxic conditions for survival [4].

Hypoxia-induced drug resistance (HDR) increases the probability of tumor recurrence and metastasis and is therefore is a critical issue in cancer therapy [5]. HDR is mainly attributed to the inhibition of drug delivery, uptake, and efficacy. The abnormal tumor vasculature in hypoxic tumors further reduces the delivery and penetration of chemotherapeutic drugs [6]. Even when drugs are taken up into tumor cells, the expression of HIF-1α promotes drug efflux via activation of the drug efflux pump, membrane-resident P-glycoprotein [7]. Moreover, the hypoxic tumor microenvironment changes the cellular proteome and genome to impede the pharmacological action of drugs, slow down/arrest the cell cycle, and block the apoptotic genes, thus decreasing the anti-tumor effect during chemotherapy [8]. Suppression of reactive oxygen species (ROS) generation under conditions lacking O_2_ also contributes to the inhibition of apoptosis [9]. Hyperbaric oxygen therapy has been presented as a systemic method to improve intratumoral oxygen partial pressure (pO_2_) and reduce HDR in radiotherapy or chemotherapy [10,11]. In addition, methods for local oxygen therapy mediated by O_2_ donors or carriers have been developed to avoid the systemic side effects of hyperbaric oxygen therapy [12].

Ultrasound (US) imaging is one of the frontline diagnostic tools in the clinic. Microbubbles (MBs) are used as an US contrast agent to provide functional information in high-contrast microvascular perfusion imaging [13]. In addition to applications in diagnostic imaging, MBs can also be used to carry therapeutic drugs, genes, or gas for local release triggered by focused US stimulation [14,15]. MBs loaded with O_2_ for local O_2_ therapy have been reported to modulate the hypoxic tumor microenvironment, inhibit HDR, and improve outcomes of chemotherapy or radiotherapy [16,17,18,19]. In addition, MB cavitation might cause various acousto–mechanical bioeffects depending on the intensity of US, such as enhanced cell permeability induced by MB stable cavitation and cell disruption induced by MB inertial cavitation [20]. MB stable cavitation produces recoverable pores on the cell membrane, a process called sonoporation, allowing direct flow of drugs into the cytoplasm and thus promoting drug uptake [21,22]. This promotion of drug uptake via enhanced permeability might be an important mechanism to overcome HDR. 

Our study focuses on the mechanisms of overcoming HDR by treatment with O_2_-loaded MBs (OMBs). Home-made OMBs combined with the chemotherapeutic drug doxorubicin (Dox) were designed for treatment of a hypoxic tumor cell model. To prevent the cytotoxicity associated with cell disruption induced by MB inertial cavitation, the optimal US parameters were defined to favor the MB stable cavitation effect. Moreover, the effective treatment dose of OMBs for reversing tumor cell hypoxia to normoxia was investigated. Finally, the improvement in drug uptake and efficacy under the HDR condition was evaluated using the predetermined optimal conditions. In addition, the relative contribution of uptake promotion and reoxygenation to overcoming HDR was investigated to determine the major therapeutic mechanism of OMB treatment (Figure 1).

## 2. Materials and Methods

### 2.1. Fabrication of OMBs

The OMBs contained 1,2-distearoyl-sn-glycero-3-phosphocholine (DSPC; Avanti Polar Lipids, Alabaster, AL, USA), 1,2-distearoyl-sn-glycero-3-phosphoethanolamine-N-[10-(trimethoxysilyl)undecanamide(polyethylene glycol)-2000] (DSPE-PEG-2000; Avanti Polar Lipids, USA), C_3_F_8_ gas, and O_2_ gas. Briefly, the lipid solution consisted of DSPC and DSPE-PEG-2000 with a weighted ratio of 10:4 in 1 mL of 5% glycerol phosphate-buffered saline (PBS). After degassing of the lipid solution, the gas mixture with volume ratio of 7:5 for C_3_F_8_:O_2_ was placed in a sealed vial. The optimal gas mixture ratio for OMB fabrication was described in our previous study [16]. Additional C_3_F_8_ gas was used to stabilize OMBs due to the hydrophilicity of O_2_. The lipid solution and gas mixture automatically formed OMBs during violent shaking by an agitator for 45 s (VIALMIX, Bristol-Myers Squibb Medical Imaging, New York, NY, USA). In addition, MBs with a pure C_3_F_8_ gaseous core were generated as controls (CMBs). The size and volume distribution of MBs were detected by a coulter counter (Multisizer 3, Beckman Coulter, Brea, CA, USA) to evaluate the quality of fabrication.

### 2.2. Evaluation of the In Vitro Stability of OMBs

An integrated platform including a US imaging system, a high-intensity focused US (HIFU) sonication system, and a real-time pO_2_ detection system was designed to monitor the contrast enhancement on US imaging and O_2_ release during HIFU-stimulated OMB cavitation (Figure 2A). A 2% agarose phantom was prepared with stick insertion to make a cylindrical hollow chamber, 7 mm diameter and 35 mm depth (total volume of 1.35 mL). A US imaging transducer was placed perpendicular to a sonication transducer and co-focused at the center of a cylindrical hollow chamber within phantom in a water tank at 37 °C. A commercial 7.5-MHz US imaging system (Terason, Model 3000, Burlington, MA, USA) was used to record US B-mode images. A 2-MHz HIFU transducer (model SU-101, Sonic Concepts, USA; focal point: 1.2 mm width × 13.3 mm depth) was triggered by a waveform generator (AWG 2041, Tektronix, Beaverton, OR, USA) and a power amplifier (Model 150A100B, AR, Souderton, PA, USA) to generate sonicated pulses for MB cavitation. A fiberoptic pO_2_ probe connected to an OxyLite 2000 system (Oxford Optronics, Abingdon, UK) was placed in the cylindrical hollow chamber within the phantom for real-time detection of the pO_2_ level in the OMB emulsion. 

The diluted OMB emulsion (1 × 10^7^ MBs/mL) was infused 1 mL into a cylindrical hollow chamber within the phantom. Since MB is a contrast agent in US imaging, the contrast enhancement of OMBs could be used to evaluate their in vitro stability. Five US images were recorded at 10-min intervals up to 60 min. The contrast enhancement of US B-mode images without HIFU sonication was measured by MatLab software (MathWorks, Natick, MA, USA). The pO_2_ level was simultaneously detected to evaluate the maintenance of O_2_ within OMBs over time.

### 2.3. Local O_2_ Release from OMBs by HIFU Sonication 

We also used an HIFU sonication system to stimulate OMB cavitation for local O_2_ release in the same integrated platform. US B-mode imaging and pO_2_ levels were monitored in real time to confirm the results. The diluted OMB emulsion (1 × 10^7^ MBs/mL) was infused into a phantom. A 2-MHz HIFU sonication system transmitted 1000-cycle pulses with 2 Hz pulse repetition frequency (PRF) to stimulate OMBs. Acoustic pressures of 1, 2, and 3 MPa were used to evaluate OMB disruption and corresponding O_2_ release. The US B-mode images were recorded at each time point. The HIFU sonication was started after image capture at 2 min and stopped before image capture at 4 min, with a total sonication time of 2 min. Finally, the enhancement of US B-mode imaging was quantified to determine the disruption threshold of OMBs. The corresponding pO_2_ levels were recorded to evaluate the efficiency of O_2_ release.

### 2.4. Cytotoxicity of Acoustic Parameters

In this study, we used transgenic adenocarcinoma mouse prostate (TRAMP) cells for the in vitro cell experiments. The cell culture medium contained 89% Dulbecco’s modified eagle medium, 10% fetal bovine serum, and 1% penicillin streptomycin. During cell culture and experimental processes, the environment of the incubator was controlled as a humidified atmosphere containing 5% CO_2_ at 37 °C. For the in vitro cell experiments, 1 × 10^4^ cells were seeded in each well of a 96-well plate for 24 h incubation.

To prevent the cytotoxicity associated with cell disruption induced by MB inertial cavitation, the cell viability was detected after OMB treatment under various HIFU parameters to define the optimal safe treatment protocol. The procedure of cell sonication is described below. After addition of OMBs to each well, culture medium was added to fill the well (volume of 400 μL/well, final concentration of 1 × 10^7^ MBs/mL) and the 96-well plate was covered by a transfer plastic membrane to prevent entry of any gas. Since MBs have the characteristic of floating, the 96-well plates were inverted to let the MBs float up and touch the cells. A 2-MHz HIFU transducer was placed upside the bottom of a 96-well plate with 55 mm interval, which was the focal depth of transducer. A water-filled plastic cone was capped on a HIFU transducer, and then transmission gel was compensated in the gap between the cone and plate. After HIFU focusing on cells, the HIFU focus (diameter of 1.2 mm) was moved manually by a triaxial platform to let the area of each well receiv one US pulse during 1 min cell sonication. In order to maintain the received number of pulses on each OMB, the HIFU sonication time was 2 min in the phantom experiments but modified to 1 min in the cell experiments. Various parameters of HIFU including PRF of 0.25, 0.5, 1, and 2 Hz (consistent acoustic pressure of 2 MPa) and acoustic pressures of 1, 2, and 3 MPa (consistent PRF of 0.25 Hz) were used to evaluate the cytotoxicity. The PRF is the number of pulse transmission in 1 s. In 1 min sonication, PRF of 0.25, 0.5, 1, and 2 Hz represents 15, 30, 60, and 120-pulse transmission. After sonication, the cell viability was measured using a cell counting Kit-8 (CCK8; Dojindo Molecular Technologies, Japan) assay with a microplate reader (Spark, Tecan, Männedorf, Switzerland).

### 2.5. Evaluation of Cell Viability and Permeability

Since irreversible sonoporation might induce cell death, the correlation between cell viability and permeability after OMB treatment was evaluated by staining with Hoechst 33342 (Invitrogen, Waltham, MA, USA) and propidium iodide (PI, Invitrogen, Waltham, MA, USA). Because the additional O_2_ supplied from OMB treatment might increase ROS generation and affect cell viability, the CMB group was included for comparison with the OMB group. The final composition of the working culture medium was 2 µg/mL Hoechst 33342 and 25 µg/mL PI. Cell sonication was performed as described above. The final concentration of CMBs and OMBs was 1 × 10^7^ MBs/mL and the predetermined optimal parameters of HIFU sonication were used (2-MHz, 2 MPa, 1000 cycles, PRF of 0.25 Hz, sonication 1 min). After treatment, fluorescence images of stained cells were captured under an inverted microscope (IX71, Olympus, Japan). Cells with nuclei stained by Hoechst 33342 (blue fluorescence) were defined as living cells. In cells with increased permeability, PI could pass through the pores in the cell membrane and into the nuclei to give red fluorescence. Cell images were analyzed by ImageJ software (NIH, Bethesda, MD, USA) to count the number of living cells and cells with enhanced permeability. The ratio of living cells to permeable cells was calculated to determine the percentage of cell permeability enhancement after MB treatment.

### 2.6. HDR Cell Model and Reoxygenation

A home-made hypoxic chamber was designed for the present study as shown in Figure 2B. The hypoxic chamber is a sealed box with two sealed membranes on the cover. One membrane was used to infuse a hypoxic gas mixture (1% O_2_, 5% carbon dioxide, and 94% nitrogen), and a 23G needle was inserted into the other membrane for placement of a fiberoptic pO_2_ probe and gas pressure balance. The hypoxic chamber was kept at a temperature of 37 °C in a heated water tank. Compared with normal tissue, the pO_2_ decreases from 30–52 mmHg to 2–32 mmHg in hypoxic tumors [23]. The real-time pO_2_ level was monitored to confirm that the chamber was maintained in a hypoxic status (<20 mmHg).

After cell seeding in a 96-well plate, hypoxia reagent (Image-iT^TM^ Red, Thermo Fisher Scientific, Waltham, MA, USA) was added to each well (final concentration of 5 µM) for 30 min incubation and then washed out and replaced with culture medium. For the hypoxic cell model, a 96-well plate was placed into the hypoxic chamber for 1–6 h incubation. The hypoxia reagent revealed red fluorescence within hypoxic cells when the oxygen level was lower than 5%. The level of hypoxia was evaluated by the red fluorescence intensity (FI) of cells by fluorescent imaging using a microplate reader. The optimal incubation time for the hypoxic chamber was defined for use in subsequent experiments.

Since OMB treatment might increase ROS generation and damage cells, the optimal OMB dose for reoxygenation of cells without further damage should be considered. Various doses of OMBs (0.5, 1.0, and 2.0 × 10^7^ MBs/mL) were added for co-incubation with hypoxic cells followed by cell sonication (2-MHz, 2 MPa, 1000 cycles, PRF of 0.25 Hz, sonication 1 min). The hypoxic level was measured before and after OMB treatment to calculate the percentage of reoxygenation in the HDR cell model, which was calculated by the ratio of the difference in red FI between normal and reoxygenated cells relative to that between normal and hypoxic cells (Figure 2C). The cell viability was also evaluated by CCK8 assay to assess possible cytotoxicity due to an overdose of OMBs.

### 2.7. Cytotoxicity and Drug Uptake after OMB Treatment

Cytotoxicity of Dox to normoxic and hypoxic cells was evaluated to demonstrate the establishment of HDR in our cell model. Cells were incubated with various doses of Dox (0, 5, 10, 15, 20, and 30 µg/well) for 4 h and then washed. The cell viability after Dox treatment was detected by CCK8 assay to determine the EC50 of Dox for subsequent experiments.

Next, Dox uptake and cell viability in the normoxia and hypoxia groups after OMB treatment were compared to investigate the effect of OMB on HDR (Figure 2D). Briefly, after seeding for 24 h, cells were placed into the hypoxic chamber for 4 h to generate the HDR cell model and were then subjected to combination treatment of Dox (LD50 = 10 μg/well) and OMB cavitation (1 × 10^7^ MBs/mL). After treatment, cells were incubated under the original culture condition (incubator for normoxia group, hypoxic chamber for hypoxia group) for 4 h and then washed. Cellular Dox uptake was evaluated by a microplate reader with excitation wavelength of 490 nm and emission wavelength of 590 nm, and cell viability was detected by the CCK8 assay.

### 2.8. Statistical Analysis

Results are presented as the mean ± standard deviation. The error bars in the graph indicate the standard deviation of each group. Differences between two individual groups and multiple groups were evaluated using Student’s *t*-test and one-way ANOVA, respectively, with a difference considered to be statistically significant when *p* < 0.05. The symbol used for the statistical significance in the graph was as follows: * *p* < 0.05 and ** *p* < 0.01.

## 3. Results

### 3.1. Size Distribution and Stability of OMBs

Two types of MBs (CMBs and OMBs) were used in this study to determine the relative contributions of uptake promotion by enhanced permeability and the reoxygenation effect by local O_2_ release to overcoming HDR. The mean diameter of CMBs and OMBs was 1.09 ± 0.04 µm and 1.14 ± 0.04 µm, respectively (Figure 3A). The volume distribution revealed the highest percentage of CMBs and OMBs with an approximate 2 µm diameter (Figure 3B). There was no significant difference in the concentration and diameter of CMBs and OMBs.

The in vitro stability of MBs is shown in Figure 3C. The US B-mode images reveal high contrast enhancement with CMBs or OMB infusion. After 60 min, CMB and OMB groups maintained a contrast enhancement of 81.2 ± 1.7% and 81.0 ± 2.8%, respectively (Figure 3D). The stability of OMBs was not significantly different from that of CMBs, demonstrating that the additional O_2_ gas in the OMB core did not influence the stability of OMBs under our optimal fabrication (*p* > 0.05). Moreover, the pO_2_ levels detected in OMBs remained constant from 0 min (162.3 ± 5.4) to 60 min (163.4 ± 1.7 mmHg), indicating the stability of O_2_ loading with insignificant leakage (Figure 3E). 

### 3.2. Local O_2_ Release from OMBs Triggered by US

After confirming the stability of OMBs, the feasibility of local O_2_ release via HIFU-stimulated OMB disruption was investigated. The real-time US imaging and pO_2_ detection system was used to observe the disruption of MBs and detect the pO_2_ level during HIFU sonication. Figure 4A shows US images at different time points to observe the change in contrast enhancement during (3 min) and after (5 min) HIFU sonication. Under acoustic pressure of 1 MPa, US images taken during HIFU sonication displayed a black hole (diameter of 3 mm) in the homogenous contrast-enhanced MB emulsion that clearly indicated the location of the HIFU focus. The US image at 5 min showed some remaining MBs. In contrast, under acoustic pressure of 2 and 3 MPa sonication, the US images showed larger black holes and non-visualization of MBs at 5 min, indicating total MB disruption after HIFU sonication.

The percentage imaging contrast enhancement after HIFU sonication with acoustic pressure of 1, 2, and 3 MPa decreased to 69.6 ± 5.2, 2.5 ± 2.1, and 1.9 ± 0.5%, respectively, in the CMB group and to 62.7 ± 5.9, 3.0 ± 2.4, and 2.6 ± 1.2% in the OMB group (Figure 4B). There was not significantly different between CMB and OMB disruption under the same acoustic pressure. Since the efficiency of MB disruption is directly proportional to the acoustic pressures, the non-significant difference in imaging contrast between the 2 MPa and 3 MPa (*p* = 0.567) indicated the complete disruption of CMBs and OMBs. The corresponding pO_2_ levels in the CMB group showed no significant increase (Figure 4C). The degree of OMB disruption increased with higher acoustic pressure, leading to an increase in pO_2_ levels. The peak pO_2_ level at the time point of 3 min increased by 3.6 ± 1.2, 7.6 ± 2.5, 8.8 ± 1.7 mmHg under 1 MPa, 2 MPa, and 3 MPa sonication, respectively. The enhanced pO_2_ levels under 2 MPa and 3 MPa were not significantly different at each time point but were significantly higher than those under 1 MPa at 3 and 4 min (*p* < 0.05). The results indicated that the disruption of 9.7 × 10^6^ OMBs increased by 7.6 ± 2.5 mmHg in pO_2_ under HIFU sonication with an acoustic pressure of 2 MPa.

### 3.3. Optimal HIFU Parameters for Cell Sonication

To prevent excessive cell damage due to MB cavitation and ROS generation, cell viability was measured to determine the optimal HIFU parameters for cell sonication. The HIFU parameters of PRF and acoustic pressure were investigated. The PRF indicates the number of US pulses per second; PRF of 0.25, 0.5, 1, and 2 Hz represents 15, 30, 60, and 120-pulse transmission. Cell viability decreased with an increase in the number of US pulses and acoustic pressure (Figure 5A,B). Cell viability was 99.4 ± 0.8, 95.0 ± 2.6, 89.1 ± 2.8, and 81.6 ± 1.6 % under 15, 30, 60, and 120 pulses, respectively, and showed no significant decrease under 15 pulses (PRF of 0.25 Hz). Moreover, significant cell damage (viability of 72.5 ± 2.9%) was induced when the acoustic pressure was higher than 3 MPa. The HIFU parameters with PRF of 0.25 Hz and acoustic pressure of 2 MPa revealed non-significant cell damage, which were suitable for the present study to investigate the cytotoxicity from chemotherapeutic drugs not MB cavitation. Thus, the optimal and safety HIFU parameters for cell sonication were set at a PRF of 0.25 Hz and acoustic pressure of 2 MPa.

### 3.4. Enhancement of Cell Permeability by OMB Treatment

The efficiency of cell permeability enhancement via MB cavitation under the optimal HIFU parameters was evaluated to improve drug uptake under the HDR condition. The fluorescent images show live cells in blue (Hoechst 33342) and permeability-enhanced cells in red (PI) (Figure 5C). The number of cells that stained red increased in both CMB and OMB groups treated with US. Quantification of the results was achieved by analysis of the fluorescent images using ImageJ. The mean number of live cells in the field of view was 483 ± 35 (each group *p* > 0.05; Figure 5D). The percentage of permeable cells increased to 20.5 ± 1.8 and 22.8 ± 5.1% in the CMB + US and OMB + US groups, respectively (each *p* < 0.01). The cell permeability enhanced by MB cavitation was not significantly different between CMB and OMB, which means the release of O_2_ would not influence cell permeability. These results show that MB cavitation could induce a local release of O_2_ and enhance the permeability of cells without cell damage under the optimal HIFU parameters for cell sonication. 

### 3.5. Optimal OMB Dose for HDR Cell Reoxygenation

Establishment of the HDR cell model was evaluated by staining with the hypoxic reagent (Figure 6A). Images show the gradual enhancement of red fluorescence in cells over time, indicating that the cells became hypoxic. The FI of hypoxic staining was directly proportional to the hypoxic incubation time and reached saturation after 4 h incubation (Figure 6B). The hypoxia level was not significantly different between the incubation time for 4 and 6 h (*p* > 0.05) to indicate the stable hypoxic status of tumor cells. Thus, the optimal hypoxic incubation time was 4 h for the subsequent HDR cell model establishment. The optimal dose of OMB for hypoxia recovery was defined by the cell viability and percent reoxygenation. Cell images show a decrease in red fluorescence with increasing OMB dose, demonstrating reoxygenation of HDR cells after OMB treatment (Figure 6C). Since an overdose of OMB might increase the production of ROS and induce cell damage, the correlation between the cell viability and reoxygenation was examined (Figure 6D). The percentage reoxygenation was 6.6 ± 6.4, 24.5 ± 12.4, 61.4 ± 14.3, and 82.0 ± 4.2% for OMB doses of 0.0 (US only), 0.5, 1.0, and 2.0 × 10^7^ OMB/mL, respectively. However, cell viability was significantly reduced to 44.2 ± 3.8% under the OMB dose of 2.0 × 10^7^ OMB/mL. The degree of MB cavitation is directly proportional to the MB doses, acoustic pressures, and pulse duration [24,25]. During MB cavitation, the high temperature and pressure could split water molecules or induce chemical reactions to generate ROS [26]. The violent MB cavitation produces mechanical force and ROS to cause various bioeffects, such as blood hemolysis [27], vascular disruption [28], and sonodynamic cytotoxicity [29]. In our study, the cytotoxicity in the HDR cell model was investigated by using chemotherapeutic drug Dox not MB cavitation. The optimal dose of OMBs for HDR cell reoxygenation was therefore determined to be 1.0 × 10^7^ OMB/mL.

### 3.6. Overcoming HDR by OMB Treatment

Measurement of cell viability with increasing concentrations of Dox confirmed HDR to Dox treatment in our HDR cell model (Figure 7A). The results showed cell viability of 90 ± 3% in the hypoxia group without Dox treatment, which indicated the process of building the HDR cell model lost 10 ± 3% live cells. The EC50 for Dox in the normoxia group (10 μg) only decreased cell viability by 25.8 ± 0.7% in the hypoxia group. The significantly higher cell viability in the hypoxia group than that in the normoxia group demonstrated the occurrence of drug resistance in the HDR cell model under different doses of Dox treatment (each *p* < 0.01). Cellular Dox uptake decreased by 6.9 ± 1.4% in the HDR cells relative to the normoxic cells (Figure 7B). The activation of P-glycoprotein on the membrane of HDR cells induces drug efflux to inhibit the intracellular drug accumulation and treatment outcome. After cell sonication, MB cavitation-enhanced cell permeability promoted Dox uptake in the CMB and OMB groups to 113.9 ± 4.4 and 115.3 ± 4.1%, respectively, in normoxic cells and to 109.3 ± 3.8 and 106.5 ± 1.7% in HDR cells. The non-significant difference in cellular Dox uptake between the CMB and OMB cavitation presented consistent outcomes with the cell permeability results to further exclude the correlation between drug uptake and present O_2_ supply in cells. These results demonstrate that the enhancement of drug uptake in HDR cells via MB cavitation-enhanced cell permeability was 14.8 ± 4.2%.

The final treatment outcome for cell viability is presented in Figure 7C. Under the normoxic condition, cell viability decreased to 50.7 ± 8.1% after Dox treatment, which is consistent with the treatment outcome of EC50. Due to the promotion of Dox uptake, cell viability was further reduced to 32.4 ± 5.5 and 36.0 ± 4.4% in the Dox + CMB + US and Dox + OMB + US groups, respectively, and showed no significant difference between groups. Under the hypoxic condition, the cell viability of each group was normalized to the control group under the normoxia condition. Cell viability decreased to 82.1 ± 4.1, 71.7 ± 4.8, and 41.6 ± 5.8% in the Dox, Dox + CMB + US, and Dox + OMB + US groups, respectively. In a comparison of the Dox and Dox + CMB + US groups, the cell viability was slightly reduced due to the promotion of cellular Dox uptake (*p* < 0.047). Although CMB cavitation enhanced cellular Dox uptake, the pharmacological cytotoxicity of Dox might be inhibited by cell hypoxia. On the other hand, the Dox + OMB + US group presented a significant decrease in cell viability relative to the Dox group (*p* < 0.01), which contributed not only the promotion of Dox uptake but also the supply of O_2_ to activate the cytotoxicity of Dox. Under the Dox treatment, cell viability was significantly higher under the hypoxic condition than the normoxia condition because of the HDR effect (each *p* < 0.01), but showed a non-significant difference in the Dox + OMB + US group (*p* > 0.05). Thus, combining Dox and OMB cavitation could overcome the HDR to improve both cellular uptake and pharmacological cytotoxicity of drugs. 

## 4. Discussion

Tumor hypoxia has always been a crucial issue in cancer treatment because the hypoxic microenvironment allows tumors to become more aggressive through chemoresistance, recurrence, and metastasis. Various O_2_ carriers or donors have been designed for local delivery or release of O_2_ at hypoxic cells or tumors for recovery of normoxia [30,31,32]. Investigating the biomechanisms in hypoxic cells after delivery of therapeutic O_2_ is important for understanding the HDR effect. 

Our study focused on in vitro HDR cell experiments to evaluate the main mechanism of OMB treatment through a comparison of reoxygenation and promotion of uptake. The acoustic characteristics of OMBs were assessed to define the optimal HIFU parameters for local O_2_ release. An acoustic pressure of 2 MPa could totally disrupt OMBs and significantly increase the pO_2_ level in the sample buffer. However, the final increase in the pO_2_ level after HIFU sonication was 2.1 ± 1.0, 4.0 ± 3.0, and 5.1 ± 1.5 mmHg in the 1, 2, and 3 MPa groups, respectively, which was lower than that achieved during the sonication. In the present in vitro experimental set up, the released O_2_ gas from OMB disruption might escape into the air without totally dissolving in the sample buffer. Although the in vitro enhancement of pO_2_ level presented lower O_2_ release, the results of cell experiments demonstrated that the optimal treatment dose of OMBs (1 × 10^7^ OMBs) recovered hypoxic cells to normoxic cells. That means an increase of 7.6 ± 2.5 mmHg in the pO_2_ level via 1 × 10^7^ OMB disruption could allow hypoxic cells to reoxygenate and inhibit HDR. The local O_2_ release and enhanced vessel/cell permeability induced by OMB treatment might improve the accumulation and diffusion of O_2_ within tumors to recover the hypoxic status [19,33,34]. Since the present study used static conditions in the in vitro experiment, the efficiency of OMB destruction was directly proportional to the number of US pulses [35]. However, the in vivo microenvironment should be a dynamic condition, and the correlation between the blood velocity and PRF plays an important role to determine the number of received pulses at a single MB in blood perfusion. Moreover, the intermittent sonication for MB refilling also influences the efficiency of the MB cavitation effect at the target regions [36]. Thus, a closed in vitro circulation system should be designed to evaluate the optimal HIFU parameters for in vivo studies and detect the actual pO_2_ level during OMB treatment.

The mechanisms of Dox to kill tumor cells include intercalation into DNA to inhibit DNA repair and ROS generation to damage cells [37]. The HDR of Dox in prostate cancer is caused by the inhibition of Dox uptake due to the expression of P-glycoprotein, the decrease of tumor cells at S-phase of the cell cycle, and the reduction of ROS generation from the absent O_2_ condition [38,39]. In our HDR cell model, HDR decreased the cytotoxicity of Dox 31.5 ± 9.1% to inhibit the treatment outcome relative to the normoxic cell model. Although the permeability of HDR cells was enhanced by MB cavitation, the 14.8 ± 4.2% increase in cellular Dox uptake only increased cell death by 10.7 ± 7.1% (Dox + CMB vs. Dox + CMB + US). With a combination of Dox and OMB treatment, the local O_2_ therapy and enhanced cell permeability improved the cytotoxicity of Dox 39.0 ± 6.3% in HDR cells (Dox + OMB vs. Dox + OMB + US). The non-significant difference in cell viability between the hypoxia and normoxia groups indicated that the treatment efficacy of Dox was recovered after OMB treatment. The respective contribution of the promotion of uptake and reoxygenation to overcoming HDR was 10.7 ± 7.1% and 28.3 ± 9.5%, respectively. In addition, protein expression and metabolism in hypoxic tumor cells might change after reoxygenation to assist the pharmacological action of chemotherapy. Sun et al. used O_2_ and paclitaxel-loaded MBs to inhibit the expression of HIF-1α, multidrug resistance-1, and P-glycoprotein in hypoxic ovarian cancer cells and improve apoptosis [40]. Khan et al. reported that cellular H_2_O_2_ levels in hypoxic MDA-MB-231 breast cancer and HeLa cells were increased by Dox-loaded O_2_ nanobubble treatment to decrease cell viability [18]. In addition, the enhancement of ROS generation for cell damage via OMB treatment was also used to combine photo/sonodynamic therapy in tumor therapy [29,41].

Although the feasibility of reoxygenation in hypoxic tumors via OMB treatment was proved, its efficacy for overcoming HDR should be studied further. Eisenbrey et al. measured in vivo pO_2_ levels within tumors and reported approximately 20–30 mmHg enhancement after OMB treatment with an OMB dose of 6.5 × 10^6^ per mouse [19,33]. In addition, our previous study showed that the intratumoral pO_2_ level increased by approximately 6 mmHg during OMB treatment with a dose of 2.0 × 10^7^ per mouse [16]. The increase in pO_2_ level changed the hypoxic tumor microenvironment to accomplish vascular normalization for improved drug delivery. Since various tumor models, microenvironments, and drugs might show different degrees of HDR, a suitable HDR-tumor model would be the main precondition to demonstrate the efficacy of OMB treatment [42,43]. The in vivo tumor hypoxia will be evaluated via immunohistochemistry of HIF-1α or pimonidazole staining to demonstrate hypoxia recovery after OMB treatment. In addition, the intratumoral gene expression of HIF-1α, VEGF, P-glycoprotein, etc., could also be detected by enzyme-linked immunosorbent assay, western blot, or real-time quantitative polymerase chain reaction to further estimate the modulation of microenvironment after tumor reoxygenation. Furthermore, the bioeffect on normal organs should be considered to prevent ROS-induced damage under the additional O_2_ supply [44,45].

## 5. Conclusions

Therapeutic O_2_ delivery has been widely applied in combination with chemotherapy to overcome the HDR effect via reoxygenation. OMB cavitation triggered by US not only releases O_2_ from the disrupted OMBs but also promotes cellular drug uptake through the sonoporation effect. However, our study demonstrated that reoxygenation had a greater contribution to overcoming HDR than the promotion of uptake. Improvement of cell metabolism and drug cytotoxicity under the normoxia condition is a crucial key to inhibition of HDR and enhancement of the treatment outcome.

## Figures and Tables

**Figure 1 pharmaceutics-14-00902-f001:**
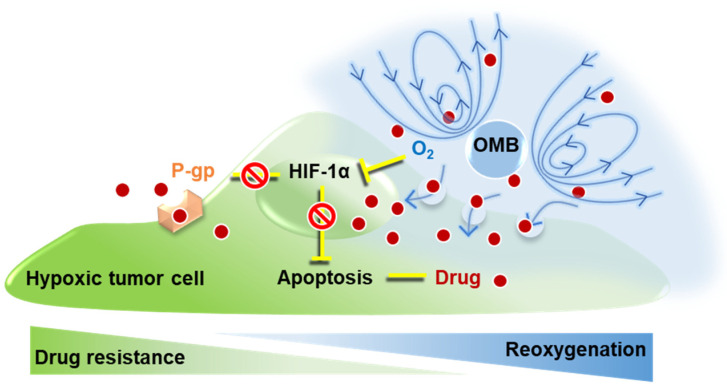
The mechanism of overcoming HDR by OMB treatment via promotion of drug uptake and reoxygenation for the in vitro cell study.

**Figure 2 pharmaceutics-14-00902-f002:**
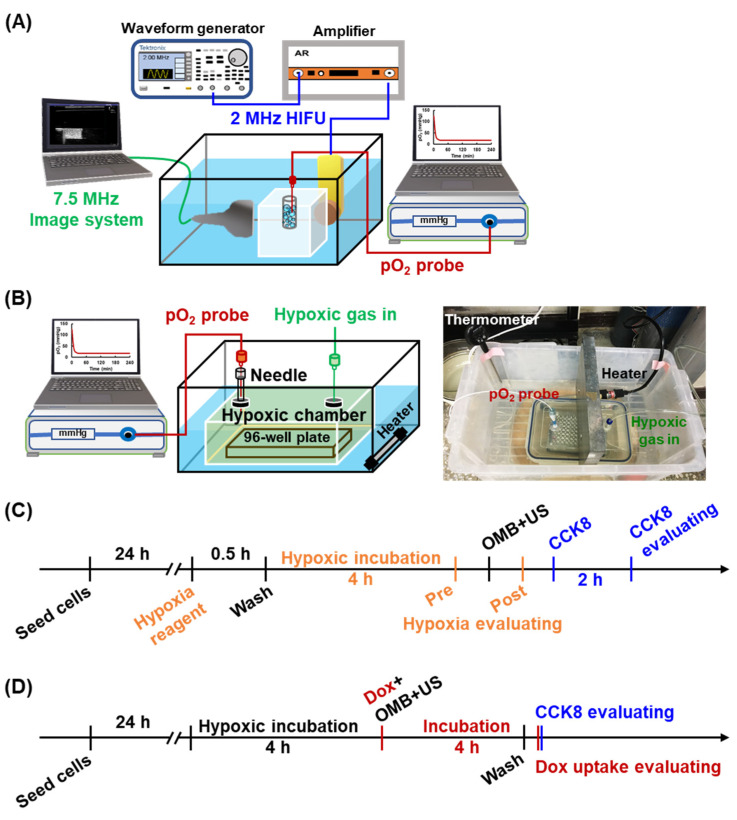
The experimental set up of (**A**) the integrated platform including a 7.5 MHz US imaging system, a 2-MHz HIFU sonication system, and a pO_2_ detection system and (**B**) the home-made hypoxic chamber for generation of the HDR cell model. Flow chart of in vitro cell experiments for (**C**) evaluation of reoxygenation and (**D**) effect of OMB treatment on HDR.

**Figure 3 pharmaceutics-14-00902-f003:**
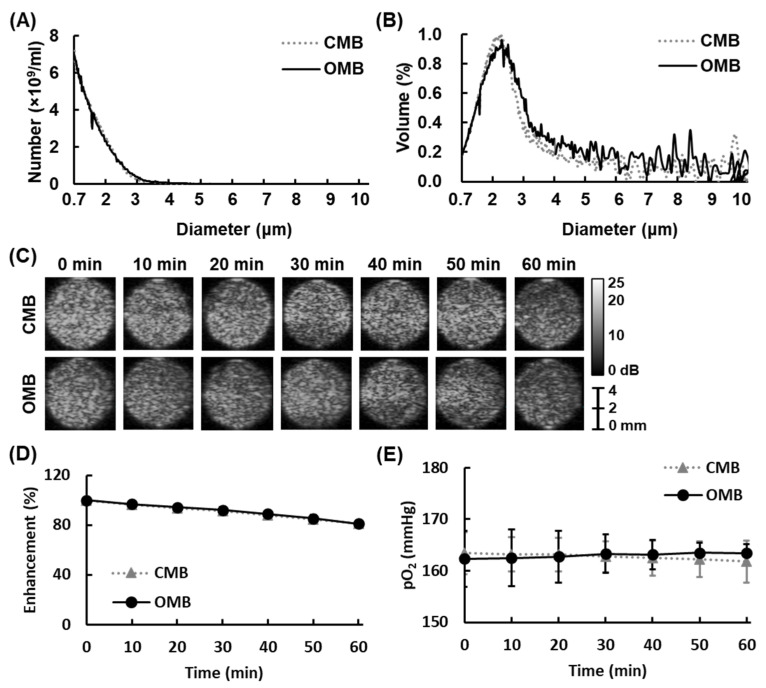
Characterization of OMBs. The size distribution of OMBs according to the (**A**) number and (**B**) volume percentage. Evaluation of OMB stability by (**C**) US imaging, (**D**) the corresponding quantification data, and (**E**) the pO_2_ level.

**Figure 4 pharmaceutics-14-00902-f004:**
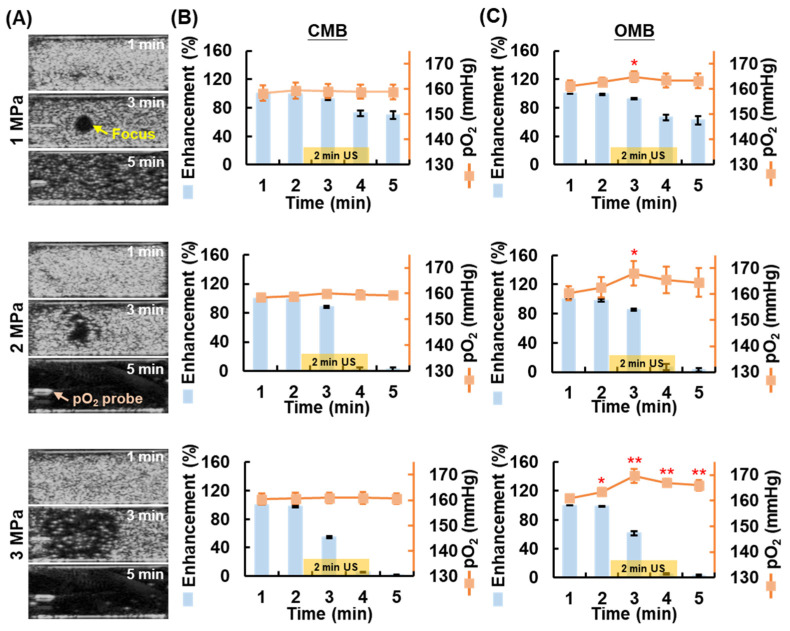
Local O_2_ release from OMB disruption under HIFU sonication with various acoustic pressures (1, 2, and 3 MPa). (**A**) US imaging reveals OMB disruption at the focal position of HIFU (yellow arrow) during sonication (at 3 min), with non-contrast enhancement at 5 min after sonication under the acoustic pressure of 2 and 3 MPa, indicating total disruption of OMBs. The position of the pO_2_ probe is indicated. The percentage of imaging contrast enhancement and corresponding pO_2_ level during HIFU sonication in the (**B**) CMB and (**C**) OMB groups. The yellow blocks indicate the period of HIFU sonication. Notably, the US images at 2 min were captured before initiation of HIFU sonication (* *p* < 0.05, ** *p* < 0.01).

**Figure 5 pharmaceutics-14-00902-f005:**
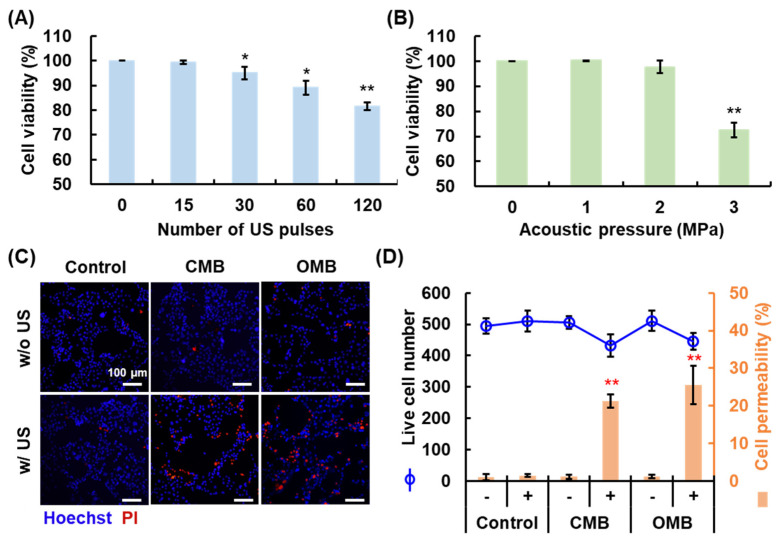
Evaluation of optimal HIFU parameters for cell sonication by cell viability under the various (**A**) number of US pulses and (**B**) acoustic pressures (* *p* < 0.05, ** *p* < 0.01). (**C**) The correlation between cell viability and permeability enhancement by OMB treatment. The fluorescence images show live cells in blue (Hoechst 33342) and permeability-enhanced cells in red (PI). (**D**) Corresponding quantification data of live cell number and the percentage of permeable cells (** *p* < 0.01).

**Figure 6 pharmaceutics-14-00902-f006:**
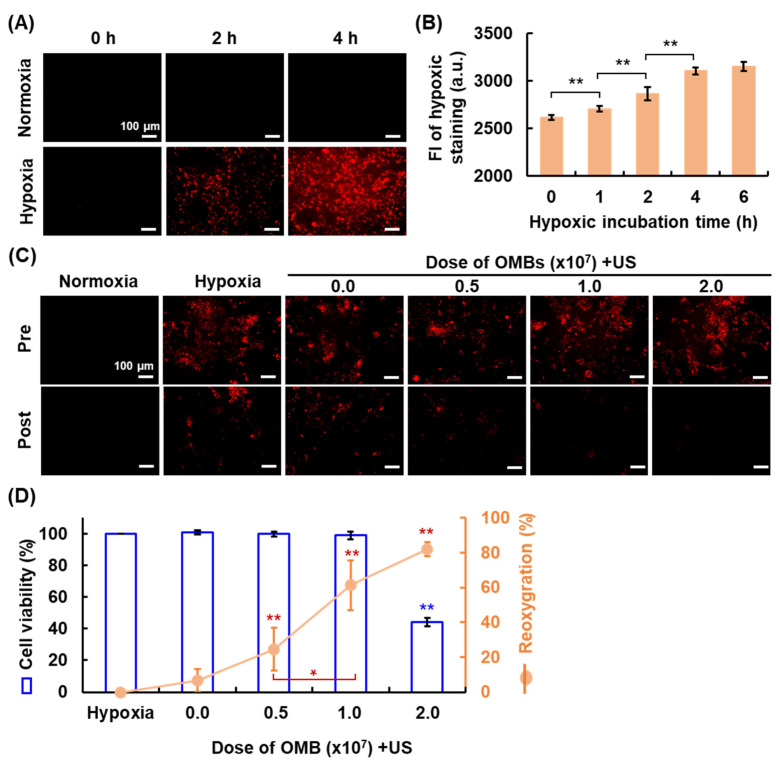
The hypoxic tumor cell model and reoxygenation. (**A**) Fluorescent images reveal HDR cells stained red by Image-iT^TM^ Red hypoxia reagent. (**B**) The corresponding quantification data for FI on cell imaging (** *p* < 0.01). The optimal dose of OMB for cell reoxygenation after OMB treatment was determined by (**C**) the fluorescent cell imaging and (**D**) the correlation between cell viability and the percentage of reoxygenation (* *p* < 0.05, ** *p* < 0.01).

**Figure 7 pharmaceutics-14-00902-f007:**
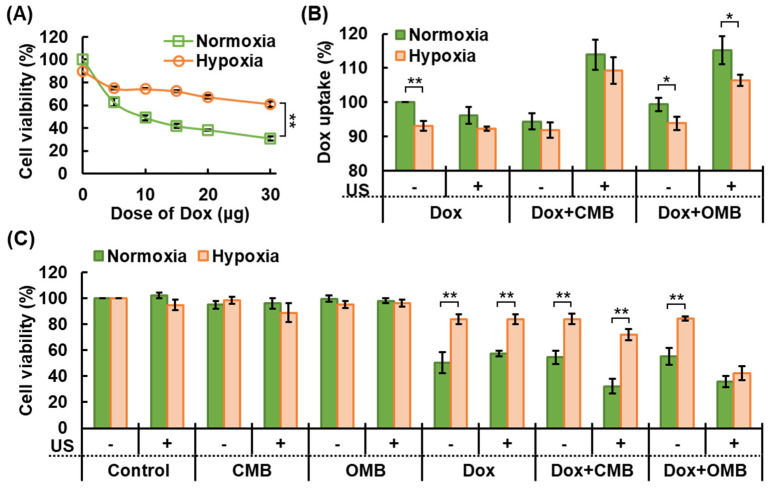
Overcoming HDR by OMB treatment. (**A**) Cell viability after Dox treatment shows the HDR effect in hypoxic tumor cells (** *p* < 0.01). (**B**) Intracellular Dox uptake was promoted by enhanced cell membrane permeability induced by MB cavitation (* *p* < 0.05, ** *p* < 0.01). (**C**) Evaluation of cell viability demonstrated that the combination of Dox and OMB treatment enhanced cytotoxicity and overcame the HDR (** *p* < 0.01).

## Data Availability

Not applicable.

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
