# Peer review of "Overcoming Hypoxia-Induced Drug Resistance via Promotion of Drug Uptake and Reoxygenation by Acousto–Mechanical Oxygen Delivery"

_pharmaceutics, 2022, doi:10.3390/pharmaceutics14050902_

Round 1
Reviewer 1 Report
It is an exciting work that deals with a subject that is not new but still relevant. Microbubbles, generally used as ultrasound contrast agents, can be used as drug/gas carriers increasing the drug uptake and reducing tumor hypoxia. Conveying these oxygen microbubbles into the tumor would reduce the problem of tumor hypoxia, which is due to the fact that tumor cells grow fast and without an organized structure. Patients with hypoxic tumors often have a poor prognosis. Hypoxia facilitates the formation of dangerous circulating tumor cells, which are the responsible cells for metastases formation, promote the proliferation of cancer stem cells, and increase chemoresistance and radioresistance. Finding a solution to reduce tumor hypoxia and increase the drug uptake capacity of hypoxic cells is undoubtedly an outstanding achievement.
In this work, the authors clearly show how loading microbubbles with oxygen can reduce cell hypoxia and increase cell drug uptake. However, as also the authors conclude, this second point seems less critical considering the results obtained.
My opinion is that the work must be accepted for publication. I would only make some small changes.
-I would add some info more in the introduction part. Info about tumor hypoxia and the effects of hypoxia on cancer cells.
-I would add a picture of the setup or at least a picture of the hypoxic chamber.
-I would change figure 6. The resolution, in my opinion, is too low.
Thank you for this interesting manuscript.
Reviewer 2 Report
The manuscript describes in vitro experiments on a therapeutic approach in which O2 carrying microbubbles (OMB) are used to improve the oxygenation status of tumors and by this increase the cytotoxicity of doxorubicin chemotherapy. In their study the authors describe phantom and cell culture experiments for characterization of the OMBs, the oxygen release by ultrasound, direct cytotoxic effects by the MB cavitation as well as the cellular uptake and cytotoxicity of doxorubicin in combination with OMB incubation. Since tumor hypoxia is a severe problem in clinical oncology limiting the efficacy of numerous O2-dependent treatment modalities, the study addresses a clinically relevant question. The manuscript is largely clearly written and the argumentation by the authors can be easily followed.
However, some relevant aspects have not been considered and the manuscript would benefit from discussing these issues.
The most import problem of the study is the amount of oxygen carried by the MBs and how representative the in vitro experiments are for the in vivo situation. From the description of the geometry of the bubbles it is possible to calculate their volume which is about 4-5x10-12 ml. Taking into account that 5/12 of the volume is oxygen (neglecting the volume of the lipids) the O2 content of one bubble is about 2x10-12 ml. The authors use an emulsion of 1x107 MBs/ml leading to an oxygen content of 0,02 µl O2/ml emulsion. This is a very, very low amount compared with arterial blood which contains 200 µl/ml blood. The emulsion in the study contains only 1:10000 the oxygen content of blood.
In their in vitro experiments the authors found a slight increase of the oxygen partial pressure pO2 by the OMB by approx. 10 mmHg. However, the pO2 does not provide any information about the O2 content. Due to Henry's law the content depends on the solubility, and the O2 solubility in water is rather low. The increase by 10 mmHg corresponds to a very low amount of oxygen. The authors should use additional methods to measure the O2 content in addition to the pO2 to provide information how much oxygen can be delivered by OMBs.
In the experiments on the O2 release (Fig. 4) the authors describe qualitatively the destruction of the OMB (change in contrast enhancement). However, it remains unclear whether the MB are destroyed completely. The authors should count the remaining intact bubbles after sonication with different parameters.
The technique the authors use to determine reoxygenation is not very suitable to assess the in vivo situation. Image-iT Red gives a signal below 5% of oxygen which corresponds to 38 mmHg which is, however, not hypoxic but the normal pO2 in many tissues (e.g., liver, subcutis). The authors should use a different method which indicates pO2<1% (»7.5 mmHg) which is a value where HIF1 is stabilized or O2-sensitive treatments become less efficient (e.g., radiotherapy).
The author should describe how they performed the sonication of a 96 well plate, especially how they focused the US pulse.
Line 32: The sentence "Hypoxic cells secrete hypoxia inducible factor-1 alpha …" is not correct because HIF is an intracellular signaling molecule which is not secrete into the extracellular space.
Line 358: the authors mention that HDR reduces the cytotoxicity of doxorubicin. The authors should explain by which mechanism of chemoresistance oxygen affects the chemoresistance against this drug in TRAMP cells (e.g., inhibition of proliferation, multidrug resistance transporters, cell-cycle arrest).
Fig. 1 is somehow misleading. It should be marked more clearly that the OMBs stay preferentially within the vessels and even with the new technique the oxygen has to diffuse from the vessel to the cell.
In Fig. 4C all pO2 curves start (t=1 min) at 160 mmHg. However, in Fig. 3E OMB showed a value of 170 mmHg. This difference is important since the increase by sonication was only 5-10 mmHg (Fig. 4C).
In Fig. 5A it is unclear whether sonication in all experiments was 2 min. If this was the case the total number of US pulses increased with higher frequency. So maybe the results shown in Fig. 5A do not correspond to the frequency but to the absolute number of pulses applied. Here the authors should perform an additional experiment in which they use a low frequency (e.g., 0.25 Hz) but for a longer time (8 min) so that the number of pulses is the same as 2 min at 1 Hz. If there would be no difference, the stability of the OHB does not depend on the frequency but on the number of pulses,
In Fig. 6B (and line 305), what is the hypothesis that with 2x107 OMB/ml the cell viability goes down dramatically? This should be explained in more detail.
In Fig. 7A, why do the curves not start at the same level (100%)? Does hypoxia per se kills cells after 4 h? But for comparison (Fig. 7C) both curves should be normalized to 100% without doxorubicin.
The size (dimension, volume etc.) of the hollow chamber (line 99) should be described. Why has a agarose phantom used around the hollow chamber? If the agarose phantom is big enough, no oxygen flow out of the chamber will occur and the experiments could also be performed in a plastic tube.
If the manuscript is printed in black and white the columns in Fig. 7 are indistinguishable.
Reviewer 3 Report
The manuscript “Overcoming hypoxia-induced drug resistance via promotion of drug uptake and reoxygenation by acousto-mechanical oxygen delivery” by Yi-Ju Ho et al investigate the mechanisms of overcoming HDR via promotion of drug uptake and reoxygenation mediated by oxygen-loaded microbubbles in a hypoxic mouse prostate tumor cell model and after 4 h of hypoxia.
In my opinion, the study is interesting, but requires substantial improvements before considering for publication.
- Line 218: please report the measure unit (µm) also for the OMBs diameter.
- Please discuss in the text the statistically significance reported in figures 4C, 5A,5B, 5D, 6B, 6D, and 7.
- The symbol used for the statistic significance should be unambiguous.
- The results have to be better discussed in the text. Often the figure legends (i.e. figure 7 c) report results not in line with that is reported in the correspondent text.
Round 2
Reviewer 2 Report
The major problem of the study is still the very low amount of oxygen delivered by the MBs. Fig. R1A clearly demonstrates that the amount of oxygen released from the MB is really low. The oxygen content (even with US) is only 50% higher than that of pure PBS (and the solubility of O2 in water/PBS is very low; see Bunsen's solubility coefficient). The oxygen content carried by MBs is below 1% of the oxygen delivered by blood with hemoglobin. The increase of the pO2 in vitro does not provide information on the in vivo situation. If a closed, water-filled system is used the release of a very small amount of O2 will lead to an increase in pO2 by 10 mmHg (due to the poor O2 solubility). However in vivo, where a steady state between oxygen supply and consumption exists, this small amount of oxygen released frpm MBs will probably not increase the dynamic pO2 substantially.
The revisions the authors made concerning this aspect (lines 410-416) do not describe the limitation of this new therapeutic approach for the in vivo situation. The authors just extrapolate their results directly to real tumors. For the readers it would be very helpful to get an impression on the amount of oxygen carried and released in tumors.
Author Response
We thank for reviewer’s comments. The ability of reoxygenation in tumors via OMB treatment was proved in our previous study. The intratumoral pO2 level was increased about 6 mm Hg (16±4 to 22±6 mmHg) after OMB treatment (2.0×107 OMBs/mouse). Although the increased pO2 level was not high, the changes of tumor microenvironment demonstrated the utility of OMB treatment. The decreased expression of HIF-1α and VEGF induced tumor vascular normalization to promote tumor perfusion and drug accumulation. Thus, we believe that our OMBs have the ability to re-oxygenate tumor hypoxia and inhibit HDR. The reference of our previous study was already described in the manuscript (line 450-454).
Line 450-454: In addition, our previous study showed that the intratumoral pO2 level increased by approximately 6 mmHg during OMB treatment with a dose of 2.0×107 per mouse [16]. The increase in pO2 level changed the hypoxic tumor microenvironment to accomplish vascular normalization for improved drug delivery.
Reviewer 3 Report
I appreciate the authors' response and modification to the manuscript. Most of the comments were addressed. In my opinion, the paper is now suitable for publication.
Author Response
We thank the reviewer for the postive comments.